# Streptozotocin-Induced Hyperglycemia Is Associated with Unique Microbiome Metabolomic Signatures in Response to Ciprofloxacin Treatment

**DOI:** 10.3390/antibiotics11050585

**Published:** 2022-04-27

**Authors:** Jenna I. Wurster, Rachel L. Peterson, Peter Belenky

**Affiliations:** Department of Molecular Microbiology and Immunology, Brown University, Providence, RI 02906, USA; jenna_wurster@alumni.brown.edu (J.I.W.); rachel_peterson1@brown.edu (R.L.P.)

**Keywords:** microbiome, antibiotics, metabolism, metabolomics, streptozotocin, hyperglycemia, ciprofloxacin

## Abstract

It is well recognized that the microbiome plays key roles in human health, and that damage to this system by, for example, antibiotic administration has detrimental effects. With this, there is collective recognition that off-target antibiotic susceptibility within the microbiome is a particularly troublesome side effect that has serious impacts on host well-being. Thus, a pressing area of research is the characterization of antibiotic susceptibility determinants within the microbiome, as understanding these mechanisms may inform the development of microbiome-protective therapeutic strategies. In particular, metabolic environment is known to play a key role in the different responses of this microbial community to antibiotics. Here, we explore the role of host dysglycemia on ciprofloxacin susceptibility in the murine cecum. We used a combination of 16S rRNA sequencing and untargeted metabolomics to characterize changes in both microbiome taxonomy and environment. We found that dysglycemia minimally impacted ciprofloxacin-associated changes in microbiome structure. However, from a metabolic perspective, host hyperglycemia was associated with significant changes in respiration, central carbon metabolism, and nucleotide synthesis-related metabolites. Together, these data suggest that host glycemia may influence microbiome function during antibiotic challenge.

## 1. Introduction

The disruption of microbiome homeostasis (dysbiosis) is a detrimental side effect of antibiotic usage [1,2]. Exposure to antimicrobial compounds rapidly and dramatically changes both the transcriptional and metabolic function of the microbiome [3,4,5]. Ultimately, antibiotic-induced dysbiosis is associated with a suite of acute and chronic negative health outcomes in humans; thus, there is a dire need for therapeutic strategies that mitigate antibiotic-related microbiome damage [6,7]. The development of these strategies will ultimately be reliant on the capacity to characterize determinants of microbiome antibiotic susceptibility.

It is well established that microbial metabolism is intrinsically linked with drug susceptibility, both in vitro and in vivo [4,8]. In fact, bacterial metabolic rate is one of the best predictors of antibiotic susceptibility [9]. Specifically, biological conditions that increase metabolic rate potentiate bactericidal antibiotics, while metabolic starvation, mutations that divert metabolism away from respiration, or environmental conditions that promote fermentation, can confer antibiotic tolerance in some species [4,8,10,11,12,13,14,15,16,17,18,19]. Within the microbiome, the fermentation of dietary fibers provides protection toward select species, suggesting that carbohydrate availability may be a key factor in the microbiome’s capacity to withstand antibiotic insult [4]. Recently, we showed that streptozotocin (STZ)-induced dysglycemia reduces fiber fermentation and increases both amino acid catabolism and primary respiration within the cecal microbiome, which results in increased susceptibility to amoxicillin [5]. To expand on this work, we used the single-dose STZ model of dysglycemia, which allows for hyperglycemic induction without dietary modification, and then we challenged mice with a short course of the fluoroquinolone antibiotic ciprofloxacin [20]. We combined 16S ribosomal RNA (rRNA) sequencing with untargeted quadrupole time-of-flight mass spectrometry (Q-TOF-MS) to profile changes in microbiome taxonomy and the cecal metabolome during ciprofloxacin treatment in hyperglycemic and normoglycemic animals. Our data demonstrate that although induced hyperglycemia does not cause dramatic restructuring of the microbiome taxonomy, it does induce significant shifts in the cecal metabolome after antibiotic treatment.

## 2. Materials and Methods

### 2.1. Animal Experiments 

All animal work was approved by the Institutional Animal Care and Use Committee (IACUC) of Brown University. Male C57BL/6J mice were purchased from The Jackson Laboratory (Bar Harbor, ME, USA) at five weeks of age, then habituated in specific-pathogen-free (SPF) conditions at 21 ± 1 °C with 12 h light/dark cycling. Animals were fed Laboratory Rodent Diet 5001 (LabDiet, St. Louis, MO, USA) and provided autoclave-sterilized water. 

After habituation, animals received an intraperitoneal injection of Na-Citrate (pH 4.5)-buffered streptozotocin (150 mg/kg) or a Na-Citrate sham (control). Animals were provided sucrose-supplemented drinking water overnight (10% for up to 18 h) to avoid hypoglycemic shock. Animals were considered hyperglycemic if they exhibited a fasting blood glucose ≥250 mg/dL when assessed 48 h post-injection using the CONTOUR^®^NEXT commercial blood glucose monitoring system (Bayer AG, Whippany, NJ, USA). Hyperglycemic and normoglycemic mice then received ciprofloxacin (12.5 mg/kg) or a pH-adjusted vehicle ad libitum via drinking water for 24 h. After this time frame, animals were sacrificed and cecal contents were harvested for taxonomic and metabolomic screening (Figure 1A). It is important to note that the animal experiments described here were performed alongside those described in recent work by Wurster et al. [5]. Specifically, the animals receiving a sham antibiotic (vehicle) were shared between these two studies. This work utilized a total of 43 mice, representing groups of 8–12 animals per experimental condition over two independent experiments.

### 2.2. 16S rRNA Amplicon Sequencing: Library Generation

Total DNA was isolated using the ZymoBIOMICS DNA/RNA Miniprep Kit (Zymo Research, Irvine, CA, USA). Amplicon sequencing libraries were generated by amplifying the V4 hypervariable region of the 16S rRNA gene, using the Earth Microbiome Project primers 515F and 806R in conjunction with Phusion high-fidelity polymerase [21,22]. The cycling protocol for amplicon generation was as follows: (1) initial denaturation at 98 °C for 30 s, (2) 34 cycles consisting of denaturation for 10 s at 98 °C, annealing at 57 °C for 30 s and extension at 72 °C for 30 s, followed by a (3) final extension at 72 °C for 5 min. Libraries were cleaned using the NucleoSpin PCR Cleanup Kit (Machery-Nagel, Düren, Germany) before being submitted to the Rhode Island Genomics and Sequencing Center at the University of Rhode Island (Kingston, RI, USA). Samples were pair-end sequenced (2 × 250 bp) on the Illumina MiSeq platform using the 500-cycle kit with standard protocols. One 16S rRNA sample was generated per animal, amplified in triplicate, and pooled. This yielded an average of 11,865 ± 6040 reads per sample. Sequencing reads were deposited to the NCBI Short Read Archive (SRA) and are publicly available as of the date of publication under the BioProject ID PRJNA811121.

### 2.3. 16S rRNA Amplicon Sequencing: Read Processing and Analysis

The R (version 3.5.0) implementation of the DADA2 algorithm was used to subject raw reads to quality filtering, trimming, de-noising, and merging as previously described [5,23]. The DADA2 function assignTaxonomy was used in combination with RDP training set 18 to perform taxonomic assignment, and diversity metrics (α and β) were calculated using the phyloseq R package (version 1.24.2) [24,25]. Differential abundance testing was performed using the DESeq2 R package [26].

Metagenomic content was predicted using PICRUSt (version 2.0) with standard parameters [27]. DADA2-generated amplicon sequencing variants were phylogenetically placed using the *place_seqs.py* script within PICRUSt. Then, gene family content was predicted using the *hsp.py* script. Finally, MetaCyc pathway abundance was predicted using the *pathway_pipeline.py* script. The scripting functions within PICRUSt are dependent on the following tools: EPA-NG and gappa (*place_seqs.py*), Caster (*hsp.py*), and MinPath (*pathway_pipeline,py*) [28,29,30,31]. Linear discriminant analysis was performed on predicted MetaCyc pathway abundances, using the Galaxy implementation of the LEfSe toolkit with standard parameters [32] (https://huttenhower.sph.harvard.edu/galaxy, accessed on 5 November 2021).

### 2.4. Q-TOF-MS: Metabolite Extraction and Annotation

Total metabolites were extracted from flash-frozen cecal samples, using an LC/MS-grade acetone:isopropanol (2:1) extraction solvent as described [5]. Metabolites were sent to General Metabolics Incorporated (Boston, MA, USA) to be analyzed via flow injection time-of-flight mass spectrometry on an Agilent 6550 iFunnel Quadrupole time-of-flight Mass Spectrometer that was run in negative ion mode and equipped with a dual AJS electrospray ionization source, as previously described (Agilent, Santa Clara, CA, USA) [33]. Metabolomics was performed as fee-for-service. These metabolomics data were acquired during the same run as the data described in Wurster et al. (2021), and the specific parameters used to operate the Q-TOF-MS are described in significant detail there [5]. As described in that work, we detected a total of 714.3 ms/spectra and 9652 transients/spectra, with a mass accuracy approximating 0.001 Da [5,33].

Data processing and putative ion annotation were performed as described, using the MATLAB Bioinformatics, Statistics, and Parallel Computing toolkits, which yielded Level D annotation to both the Human Metabolome Database and KEGG [5,33,34]. Data annotation was automated using a proprietary platform at General Metabolics and delivered upon run completion. Principal Coordinates Analysis (PCoA) was performed on annotated ion intensities and statistically tested with permutational ANOVA (PERMANOVA) using the vegan R package (version 1.26.1) [25].

### 2.5. Q-TOF-MS: Computational Analysis

The DESeq2 R package (version 1.26.1) was used to perform differential abundance testing on ion intensities [26]. Differentially abundant metabolites (Benjamini–Hochberg adjusted *p*-value < 0.05) that included a KEGG annotation were subjected to pathway enrichment analysis using the PAPi R package [35,36]. Only pathways with an adjusted *p*-value < 0.05 were considered statistically significant. 

## 3. Results

Following a 2-week habituation period, 7-week-old male C57BL/6J mice were given a single intraperitoneal injection of either STZ or a sham vehicle. After 48 h, animals were assessed for hyperglycemia and subsequently randomized. The next day, ciprofloxacin (12.5 mg/kg) or a vehicle control was administered ad libitum via the drinking water for 24 h before animals were sacrificed and their cecal contents were collected for taxonomic profiling and untargeted metabolomics (Figure 1A). This antibiotic concentration was selected based on past work indicating a capacity to perturb the microbiome within a 24 h period, and relevance to clinical concentrations [4,37].

First, we examined cecal β-diversity before and after ciprofloxacin treatment using PCoA paired with PERMANOVA (Figure 1B). In line with our previous research, both STZ-induced hyperglycemia and antibiotic treatment were associated with significant divergence in community structure (Figure 1B) [5]. Hyperglycemia was associated with a single significant taxonomic shift: the expansion of Verrucomicrobia, which has been previously attributed to the abundance of *Akkermansia muciniphila* (Figure 1C) [5]. Interestingly, hyperglycemia had no impact on the post-antibiotic expansion of Firmicutes, but did appear to exaggerate the reduction in Bacteroidetes (Figure 1C). However, upon further examination, the difference in abundance was not statistically significant (adjusted *p*-value = 0.2). To profile host-dependent differences in taxonomic composition after ciprofloxacin treatment, we next performed differential abundance testing on genus-level amplicon sequence variant (ASV) abundance [26]. Surprisingly, the abundance of very few taxa were STZ-dependent in response to ciprofloxacin, based on interactions term analysis. Hyperglycemic mice had a less severe reduction in *Clostridia_sensu_stricto* and *Parasutterella* (Figure 1D: positive interaction) and did not experience the increase in *Duncaniella* exhibited by controls in response to ciprofloxacin (Figure 1D: negative interaction).

To further profile the differences between STZ-treated and control communities after ciprofloxacin exposure, we again examined genera-level differential ASV abundances. We found that STZ and ciprofloxacin co-treatment impacted the abundance of several Firmicutes; the abundance of *Neglecta* increased, while the genera *Kineothrix*, *Eisenbergiella*, and *Acutalibacter* significantly decreased compared to normoglycemic ciprofloxacin-treated animals (Figure 1E). Because paired metagenomic and metatranscriptomic sequencing were not performed, it is impossible to make definitive claims about microbiome function in these samples. However, computational tools such as PICRUSt can be implemented to predict metagenome content from 16S data, allowing for functional inference [27,38]. Using PICRUSt2, we predicted differences in MetaCyc pathway-related gene content that were uniquely affiliated with hyperglycemic or normoglycemic mice after ciprofloxacin (Figure 1F). Strikingly, despite their similar taxonomic compositions, the predicted metagenome of the STZ and ciprofloxacin-cotreated microbiota was distinct from that of normoglycemic ciprofloxacin-treated controls. This is likely explained by the additive effect of functionally redundant pathways changing in unison without contributing to taxa-level significance. Overall, hyperglycemic communities had a greater variety of associated MetaCyc pathways, with notable enrichment in nucleotide metabolism, monosaccharide capture, menaquinone generation, aerobic respiration, and TCA cycle activity (Figure 1F). Ultimately, this suggests that the STZ-treated microbiota has more significant differences in functional potential than the normoglycemic microbiota after antibiotic exposure.

Given the host-dependent differences in predicted metagenome content, we next sought to characterize the cecal metabolome in hyperglycemic and normoglycemic mice during ciprofloxacin treatment using Q-TOF-MS (Figure 2). First, we profiled the diversity of the cecal metabolome via PCoA paired with PERMANOVA, and found that both hyperglycemia and antibiotic exposure dramatically shape the diversity of the metabolome (Figure 2A). We then performed differential abundance testing to examine which predicted Q-TOF-MS metabolites and KEGG pathways were altered after ciprofloxacin exposure, and which had altered abundances during antibiotic treatment in a host-dependent manner (Figure 2B, Appendix A). After ciprofloxacin treatment, hyperglycemic communities were enriched for metabolites involved in purine metabolism, peptidoglycan synthesis, and dietary-fiber components like isoflavonoids and phenylpropanoids (Figure 2B). Simultaneously, metabolites involved in nitrogen metabolism, carbon fixation, energy carrier generation, catabolism, pyruvate processing, and TCA activity were all depleted in STZ-treated communities relative to normoglycemic controls (Figure 2B). It is likely that the reduction in central carbon metabolites (pyruvate, TCA, energy carrier generation, etc.) reflects the diminished transcription of the affiliated pathways and represents a ciprofloxacin-specific response akin to what was identified by Cabral et al. [4,37]. The spike in purine and peptidoglycan metabolites may additionally indicate increased ciprofloxacin sensitivity in the STZ-treated microbiome, as ciprofloxacin causes lethal stalling of DNA replication that can induce the bioaccumulation of nucleotides, nucleosides, and cell wall synthesis components [39,40,41].

Supporting evidence for host-dependent microbiome function during ciprofloxacin treatment can be seen in the differentially abundant metabolites that are subjected to STZ interaction (Figure 2C). STZ-treated communities uniquely exhibit an overall increase in multiple sugars and sugar alcohols, including hexonic acids, ribitol, and pentose during ciprofloxacin treatment, suggesting that STZ-induced hyperglycemia increases cecal monosaccharide concentrations (Figure 2C). Multiple features involved in central carbon metabolism showed host-specific regulation in response to ciprofloxacin, including STZ-specific elevation in carnitine electron acceptors, nicotinate, and pyruvate oxime, and an STZ-specific decrease in isocitrate and malate [42,43] (Figure 2C). These data strongly suggest active differences in TCA cycle activity between STZ-treated and control mice after ciprofloxacin exposure. Because modifications in the TCA cycle are a characterized microbiome response to ciprofloxacin [4], differences in TCA activity may ultimately indicate host-dependent ciprofloxacin susceptibility. We additionally observed STZ-specific accumulation of nucleotide-generation metabolites, including conjugated and unconjugated uracil, inosine, and deoxyuridine (Figure 2C), which may indicate differences in ciprofloxacin-related DNA replication rates between STZ-treated and control mice.

Other metabolites that were varied between hyperglycemic and normoglycemic mice after antibiotic exposure included those involved in cholesterol metabolism, tryptophan metabolism (indole-3-acetate), heme processing (dueteroporphyrin IX), amino acid catabolism (3-dehydroshikimate, 6-methylnicotinamide, and ketovaline), and lipid processing (Figure 2C) [44,45,46]. Together, these data highlight that even after antibiotic exposure, host hyperglycemia is associated with significant changes in the metabolic environment of the cecal microbiome, which undoubtedly impacts upon microbial function and metabolic capacity.

## 4. Discussion

The key goal of this work was to expand upon the observation that streptozotocin-induced hyperglycemia increased microbiome antibiotic susceptibility to amoxicillin, by examining whether this trend held true for a structurally distinct antibiotic, ciprofloxacin. Here, we present a discovery platform that pairs 16S rRNA sequencing data with predicted metabolite abundances to yield data that may inform the synergy between host hyperglycemia and ciprofloxacin toxicity within the microbiome. In this work, we found that induced hyperglycemia impacted community diversity and drastically modified the composition of the cecal metabolome following ciprofloxacin treatment. We observed notable differences in metabolites involved in respiration, central carbon metabolism, and nucleotide metabolism, all of which are biological processes that are inherently involved in microbial ciprofloxacin susceptibility.

Transcriptional modifications of nucleotide synthesis and salvage pathways have previously been characterized as a microbial signature of ciprofloxacin susceptibility [4]. Ciprofloxacin has additionally been shown to induce significant changes in both gene and protein expression within select species, with robust alterations to central carbon metabolism, amino acid generation, and respiration being consistently reported [40,47,48]. In fact, mutations in core metabolic functions represent a primary pathway toward resistance to this antibiotic, and the stimulation of these pathways is sufficient to sensitize bacteria to ciprofloxacin [49,50]. Thus, modifications in TCA activity appear to be a conserved microbiome response to this antibiotic. Our observed differences in respiratory capacity (as indicated by predicted metagenome content and metabolite abundances) and key ciprofloxacin-responsive pathways in STZ-treated hyperglycemic and normoglycemic control mice may be indicative of divergence in microbiome function. For example, the observed increase in monosaccharide import capacity and primary respiration is congruent with the amoxicillin-specific findings presented in Wurster et al., where elevated environmental sugar levels prompted an increase in phosphotransferase import and glycolysis within the hyperglycemic microbiota [5]. However, this hypothesis cannot be fully confirmed without validated functional screening of the microbiome.

While these data have exciting implications, there are clear limitations in our experimental methods that complicate biological interpretation and prevent the use of this data set for mechanistic validation purposes. A key caveat of our metabolomic protocol is that flow injection mass spectrometry does not include the chromatography separation that is considered the gold standard of high-resolution MS within the field. Thus, it is likely that there is some ambiguity in our metabolite annotations. In order to gain unambiguous high-confidence ion annotation, further work would need to utilize both chromatography-based separation and validation against reference standards, which is a labor-intensive process that could take years, depending on the number of standards required [51]. Untargeted metabolomics (such as Q-TOF-MS) provides the advantage of reporting the relative abundance of metabolites, data that is easily integrated into other microbiome ‘omics datasets. However, it is important to acknowledge that these data are not sufficient for reporting on total metabolite abundance within samples [52]. Thus, we must interpret the data presented in this study within the context of an untargeted, discovery-based framework. Further studies will need to implement both higher-resolution machinery and targeted metabolomics to quantify metabolite levels within these complex biological samples. Another major caveat is that the untargeted metabolomics protocol used here cannot distinguish between host-derived and microbially derived compounds. Thus, alternative strategies (such as metagenomics or metatranscriptomics) will need to be implemented to confirm whether a metabolite shift is bacterial in origin [5,51,53]. Another key limitation to consider is that 16S rRNA sequencing cannot reach species- and strain-level taxonomic classification [54]. Finally, the PICRUSt2 analysis utilizes 16S rRNA ASV data and is thus equally limited by taxonomic resolution. Thus, any subsequent examination of the interplay between host metabolism and antibiotic activity within the microbiome will need to rely on methods within higher functional resolution to gain meaningful mechanistic insights.

## Figures and Tables

**Figure 1 antibiotics-11-00585-f001:**
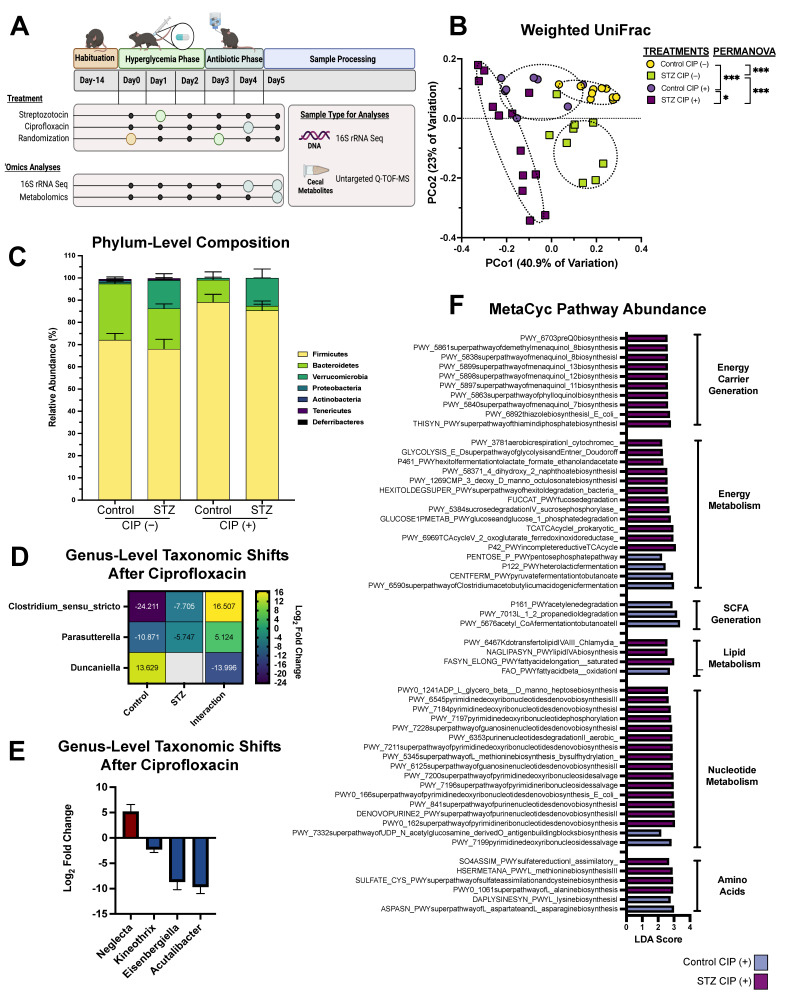
The impact of streptozotocin and ciprofloxacin treatment on microbiome composition. (**A**) Experimental design used in this study. Figure made with BioRender.com. (**B**) Weighted UniFrac Distance between 16S rRNA amplicons. (**C**) Relative abundance of detected phyla in 16S rRNA amplicons. Data represent mean ± SEM. (**D**) Differentially abundant bacterial genera following ciprofloxacin treatment in STZ-treated and normoglycemic mice versus vehicle controls alongside their interaction value. Data represent log_2_ fold change. (**E**) Differentially abundant bacterial genera between STZ-treated and control mice after ciprofloxacin administration. Data represent log_2_ fold change ± SEM. (**F**) Linear discriminant analysis of MetaCyc pathway abundance as predicted using PICRUSt. Data represent STZ-treated versus control mice after ciprofloxacin treatment. For all panels, *n* = 8–12 per group for (**B**): permutational ANOVA (* *p* < 0.05; *** *p* < 0.001). For (**D**,**E**): differentially abundant = Benjamini–Hochberg adjusted *p* value < 0.05.

**Figure 2 antibiotics-11-00585-f002:**
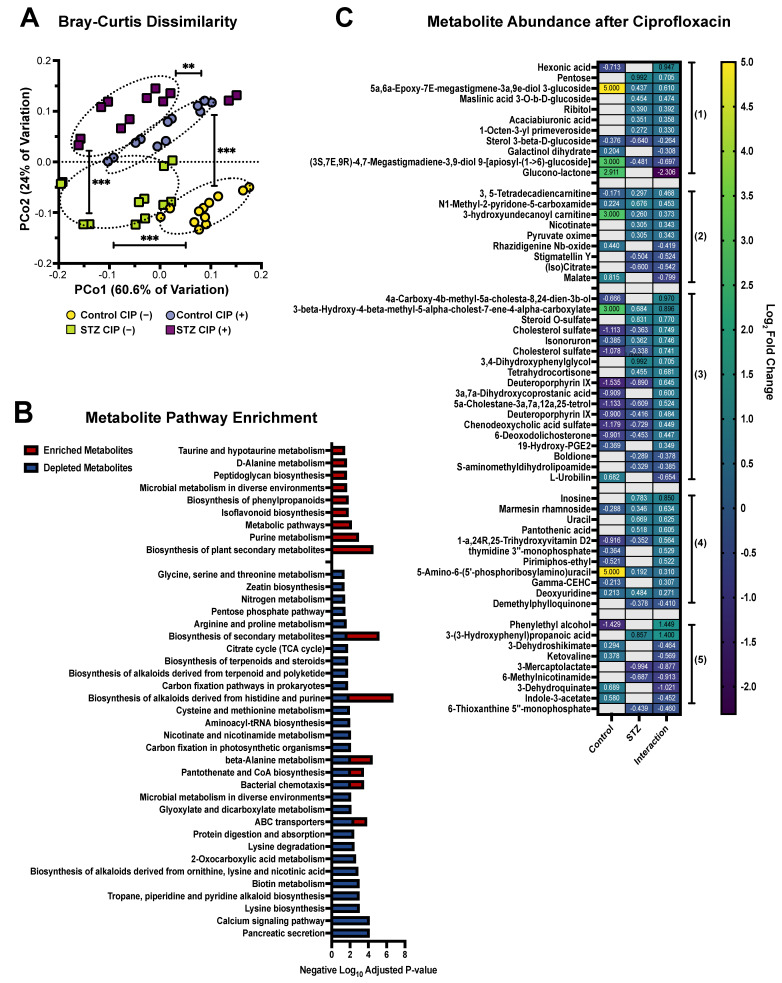
Streptozotocin-induced hyperglycemia is associated with metabolome divergence after ciprofloxacin treatment. (**A**) Bray–Curtis dissimilarity of Q-TOF-MS extracts from experimental groups. (**B**) KEGG pathway enrichment of differentially abundant Q-TOF-MS metabolites after ciprofloxacin treatment. Data represent STZ-treated mice versus normoglycemic controls. Split-colored bars indicate that this biological pathway contains metabolites that were both enriched and depleted. (**C**) Differentially abundant Q-TOF-MS metabolites in control and STZ-treated mice during ciprofloxacin treatment. Data represent log_2_ fold change versus vehicle-treated controls ± SEM. Numbers represent grouping by biological pathways: (1) monosaccharides, (2) central metabolism and respiratory metabolites, (3) steroid and heme biosynthesis and processing, (4) nucleotide metabolism, (5) amino acid metabolism. For full results see Appendix A. For all panels, *n* = 6 per group, with 2 technical replicates per group. For (**A**): permutational ANOVA (** *p* < 0.01; *** *p* < 0.001).

## Data Availability

Sequencing reads were deposited to the NCBI Short Read Archive (SRA) and are publicly available as of the date of publication under the BioProject ID PRJNA811121. Q-TOF-MS data are available in the Appendix A.

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
