# Peer review of "Streptozotocin-Induced Hyperglycemia Is Associated with Unique Microbiome Metabolomic Signatures in Response to Ciprofloxacin Treatment"

_antibiotics, 2022, doi:10.3390/antibiotics11050585_

Round 1

Reviewer 1 Report

Reviewing of Manuscript Number: Antibiotics-1639707

Title:  Streptozotocin-Induced Hyperglycemia is Associated with Unique Microbiome Metabolomic Signatures in Response to Ciprofloxacin Treatment

Summary: the proposed article deals with the impact of host dysglycemia on microbiome after ciprofloxacim treatment. The aim is to determine antibiotic susceptibility determinants thanks to combination of RNAseq and untargeted metabolomics.

Broad comments:

I stopped reviewing the manuscript after mat & meth because of a major problem with the results:

The conclusions of the authors' work are based on the pathways which are notably identified thanks to the metabolites detected. Their identification must be validated. But, for the analysis of the metabolome, the authors simply performed Q-TOF-MS analyses. Therefore, none of the metabolites can be annotated at level 1 (i.e. with a validated identification using standard). I'm also not sure that the metabolites are annotated at level 2. For that, it would be necessary to compare clean MS/MS spectra with spectral libraries. Prior LC separation is required for that. In the absence of separation by LC, it would have been necessary to perform FTICR-MS. In the other hand, my doubts about the annotation seem to be confirmed with molecules which do not ionize in negative mode in my opinion.

Specific comments:

I regret that the manuscript does not include line numbers to simplify the revision.

I regret also that experimental design is not sufficiently described and that we must repeatedly refer to the publication Wurster 2021.

Introduction

  • Introduction is well-written and sufficiently documented.

Materials and Methods

  • I’m not convinced that sentence “Principal Coordinates Analysis (PCoA) was performed on annotated….” must be placed in 2.4. paragraph.
  • Unless I am mistaken, it seems to me that the R packages phyloseq and DESeq are used for the RNAseq data and not the metabolomics data. Please clarify which data are used in which analyzes with which methods or packages and please place the information in the correct paragraphs (or rename paragraphs correctly in function of contained information).

More comments are attached within the PDF file of the manuscript.

Decision: I reject the article because the conclusions are based on the results which are not validated (putative annotation of metabolites)

Author Response

Please see the attachment. Thank you for your time and consideration. 

Reviewer 2 Report

The manuscript by Wurster and co-authors explores the effect of streptozotocin-induced hypoglycemia on the response of intestinal microbiota to ciprofloxacin treatment. Due to the fact, that hypoglycemia is a frequent pathological condition in diabetic patients, a study of the effects of this condition on changes in the gut microbiota caused by antibiotic administration is extremely important in the context of developing strategies for antibiotic use in diabetic patients. Authors use 16S metabarcoding coupled with Q-TOF-MS analysis of metabolites to assess both compositional and functional changes in the microbial communities. The main limitations of this work, for my view, is the absence of correlation analysis between the composition of microbial communities and the metabolomic data (canonical correspondence or redundancy analysis). However, in “Discussion” section the authors reasonably state in, that “untargeted metabolomics protocol used cannot distinguish between host-derived and microbially-derived compounds, thus alternative strategies will need to be implemented to confirm if a metabolite shift is bacterial in origin”. I completely agree with this, however still curious if such attempts were performed.

Nevertheless, the paper is well written, figures are fine and explanatory. The results will be of great interest to the scientific community. I would recommend accepting the paper with following minor improvements:

  1. PRJNA811121 is not publicly available at the moment, please release the data.
  2. Chapter 2.2. Authors cite the previous paper in Cell Reports to describe the library prep and sequencing methods. I would suggest adding the short summary in this paragraph (primers, which hypervariable region was used) to make it more convenient to the reader.
  3. How many 16S samples were finally analyzed? Were there any replicates?
  4. Chapter 2.3. DADA2 is ok, however methods of secondary 16S analysis (alpha & beta-diversity, PERMANOVA analysis) are not described. Please add some information about packages, which were used for this analysis.

Author Response

(The authors gave the same response as above.)

Reviewer 3 Report

I feel that the overall idea generation of the manuscript is well-conceived. The authors have proposed that streptozotocin-induced hyperglycemia can alter microbiome metabolic signatures when treated with ciprofloxacin. Ciprofloxacin is a well-known antibiotic from the fluoroquinolone group. It is widely used in various diseases, even in diabetic patients. It is well-documented that the natural microbiome plays an important role in governing human health. However, when antibiotics are consumed to treat infections, the antibiotics significantly damage the microbiome. The damage to the natural microbiome can lead to altered human physiological processes.

The authors have performed 16S rRNA sequencing and untargeted metabolomics to understand the alteration in the microbiome taxonomy and microbiome environment. The authors found that there were significant changes in the microbiome function when they were challenged with ciprofloxacin. The alteration in the microbiome function was dependent on the host glycemic levels.

I found the study well-written. The conclusions are well-supported with the results and discussions of the prior literature.

However, I have the following queries:

  • How many groups of animals were there?
  • In each group, how many animals were kept?
  • Were the mice conditioned with a placebo before the tests were carried out? If not, please explain why.

The paper can be accepted after the aforesaid minor corrections.

Author Response

Please see attachment. Thank you for your time and consideration.

Round 2

Reviewer 1 Report

I appreciate that the authors have taken into consideration my previous comments.

However, I have new some comments:

  • Figures are too small to read on paper. A zoom of 300% is needed!!
  • Figure 1C, the legend is not clear (black frame too thick) and only three kind of bacteria are visible out of seven.
  • The choice of bacteria genus studied is not clear in the result part. In general, are these bacteria particularly present in the cecum? Why the study was performed in two stages and with a different result presentation (fig 1 D & E)? Maybe it would be interesting to link these bacteria to their respective phylum.
  • Fig2C: Do the values in the data frame correspond to log(2)FC? So the SEM values are not indicated, are they? The log scale on the right is useless if the color code is not added (same for Fig 2D).
  • There is no real discussion. The authors state their results and simply add references at the end of the sentences. There is no text relating to the comparison of their results with those of the literature.
